# How do diverse low-income and middle-income countries implement primary healthcare team integration to support the delivery of comprehensive primary health care? A mixed-methods study protocol from India, Mexico and Uganda

Rohina Joshi [1,2] Innocent Besigye,[3] Ileana Heredia-Pi [4] Manushi Sharma,[2] David Peiris [5,6] Robert James Mash [7] Hortensia Reyes-Morales [8] Felicity Goodyear-Smith [9] Renu John [2] Doris V Ortega-Altamirano [4] Emanuel Orozco-Núñez [4] Leticia Ávila-Burgos,[4] Ragavi Jeyakumar,[5,6] Edson Serván-Mori,[8] Sanjeev Upadhyaya,[10] Varun Arora,[11] D Praveen[6,12]

For numbered affiliations see end of article.

**Correspondence to**
Associate Professor Rohina Joshi;
rohina.joshi@unsw.edu.au

## ABSTRACT

**Introduction** Attainment of universal health coverage is feasible via strengthened primary health systems that are comprehensive, accessible, people-centred, continuous and coordinated. Having an adequately trained, motivated and equipped primary healthcare workforce is central to the provision of comprehensive primary healthcare (CPHC). This study aims to understand PHC team integration, composition and organisation in the delivery of CPHC in India, Mexico and Uganda.

**Methods and analysis** A parallel, mixed-methods study (integration of quantitative and qualitative results) will be conducted to gain an understanding of PHC teams. Methods include: (1) Policy review on PHC team composition, organisation and expected comprehensiveness of PHC services, (2) PHC facility review using the WHO Service Availability and Readiness Assessment, and (3) PHC key informant interviews. Data will be collected from 20, 10 and 10 PHCs in India, Mexico and Uganda, respectively, and analysed using descriptive methods and thematic analysis approach. Outcomes will include an in-depth understanding of the health policies for PHC as well as understanding PHC team composition, organisation and the delivery of comprehensive PHC.

**Ethics and dissemination** Approvals have been sought from the Institutional Ethics Committee of The George Institute for Global Health, India for the Indian sites, School of Medicine Research Ethics Committee at Makerere University for the sites in Uganda and the Research, Ethics and Biosecurity Committees of the Mexican National Institute of Public Health for the sites in Mexico. Results will be shared through presentations with governments, publications in peer-reviewed journals and presentations at conferences.

## STRENGTHS AND LIMITATIONS OF THIS STUDY

⇒ This study will provide insight into the availability of policies for primary healthcare (PHC) workforce in three diverse countries.
⇒ It will help understand the implementation of policies on PHC workforce, and team organisation for the delivery of comprehensive primary healthcare.
⇒ While each country includes sites from regions representative of the health system of the country, the results are not generalisable beyond the region due to the wide variation in socio-demographic factors and health system structure.

## INTRODUCTION

Primary healthcare (PHC) in many low-income and middle-income countries (LMICs) is fragmented, selectively disease-oriented and under-resourced with suboptimal performance.[1–4] There is global recognition of the need to strengthen PHC because it is essential for all to have access to affordable high-quality healthcare which is considered the path towards achieving universal health coverage (UHC), the main target for Sustainable Development Goal (SDG) 3.[5] Appropriate high-quality PHC is considered as the most equitable and efficient way to enhance the health of populations.[5–7]

In 2017, the WHO developed a framework on integrated people-centred health services (IPCHS), which called for a fundamental shift in the funding, organisation

and management of health services.[1 8] IPCHS encourages 'people-centred', rather than 'disease focused' and 'siloed' health systems, thereby supporting the progress of countries towards UHC. High-quality PHC is people-centred, accessible, coordinated, comprehensive and continuous. PHC describes an approach to health policy and service delivery that includes both primary care services delivered to individuals, and public health services delivered to populations.[9] The delivery of high-quality PHC is dependent on the availability of adequately skilled and motivated PHC workforce, and the way in which they function as collaborative teams. Workforce availability in turn depends on the country's PHC workforce policies, funding, remuneration, supportive supervision and professionalisation.

This workforce refers to all occupations of health professionals responsible for organising and delivering PHC,[10] essential to deliver high-quality PHC services.[11–13] In a context of increasing demand for healthcare, driven by demographic, epidemiological and technological changes, the PHC workforce needs to adapt to these changes.[14]

Policies on the PHC workforce, formation of integrated PHC teams, and the capacity of these teams to deliver high-quality PHC varies between countries. It is therefore important to understand how PHC teams are organised, and whether the services delivered are truly comprehensive. We define PHC teams as a structured group of multidisciplinary health workers, co-located in a facility and serving a defined population in the community.[15] We use Barbara Starfield's definition of comprehensiveness which refers to the provision of holistic and appropriate care across a broad spectrum of health conditions, across the life span and treatment modalities.[16] While there is a body of research on PHC systems, recent reviews have indicated knowledge gaps on effective PHC team organisation and service delivery.[3 12 14 17] In particular, given variability in health system contexts, there is a need to investigate how different LMICs organise and integrate their PHC teams to deliver comprehensive care.[3 5]

Against this backdrop, the PHC Research Consortium[17] commissioned researchers from India, Mexico and Uganda to study PHC team organisation and delivery of comprehensive PHC services (see table 1). The aim of this research is to investigate the relationship between different ways of organising PHC workforce and their delivery of comprehensive PHC in three LMICs: India, Mexico and Uganda, being three large and diverse countries on different continents. Specific objectives are to:

1. Review the national and subnational policies on PHC team composition and organisation and expected comprehensiveness of PHC service delivery.
2. Describe the actual composition and organisation of PHC teams in the sampled health services.
3. Assess the comprehensiveness of care provided by these teams using the above definition.
4. Conduct a comparative analysis of the relationship between PHC team composition and organisation with

the delivery of comprehensive PHC across the three countries.

## Primary healthcare context in India, Mexico and Uganda

The definition of PHC used in our study is consistent with the Alma-Ata declaration which includes preventive, promotive, curative and palliative services available at the lower levels of the health system. This study will be conducted in the context of recent health system reforms, commitment of Ministries of Health to SDG Target 3c (Substantially increase health financing and the recruitment, development, training and retention of the health workforce in developing countries) and attainment of UHC (SDG3.8) through PHC.

PHC System in India: Comprehensive Primary Healthcare (CPHC) has always been the essence of the Indian health system policy. The health reforms of 2005 and 2017, and 2018 focused on the actionable and achievable tasks through which CPHC is being realised. In 2005, the National Health Mission, aimed to strengthen the rural health services and provide financial protection to families below the poverty line. Building on this momentum, in 2017, further reforms were made to put CPHC at the forefront. CPHC was intended to address both communicable and non-communicable diseases through PHC centres with multidisciplinary teams, and to establish new PHC facilities at the village level. These would then link to the PHC, secondary and tertiary health centres. Finally, in 2018, the Government of India introduced Ayushman Bharat (UHC) comprising two major health initiatives—Health and Wellness centres (Upgradation of existing PHCs and sub centres to provide CPHC) and Pradhan Mantri Jan Arogya Yojana (provision of health cover of Rs. 5 lakhs (~US\$7000) per family per year for secondary and tertiary care hospitalisation) covering the entire spectrum of prevention and promotion along with primary, secondary and tertiary care.[18]

PHC system in Mexico: The Mexican public healthcare sector is organised around a segmented model and is marked by the separation of healthcare rights between the insured in the salaried, formal sector of the economy and the offer of health services for the poor and uninsured, the latter organised by the recently created Health Institute for Welfare (Instituto de Salud para el Bienestar or INSABI by its Spanish acronym). All population segments receive their health services through vertically integrated institutions, each of which is responsible for stewardship, financing and service delivery only for that particular group.[6–8] For example, the Mexican Social Security Institute (by its Spanish acronym) covers the employees of the formal private sector and employees of the army are covered by the Social Security Institute for the Mexican Armed Forces. Launched in 2015 and still in the early stages of implementation, the government's Comprehensive Health Care Model (MAIS by its Spanish acronym) aims to define and monitor patients' care pathways through the system to ensure timely delivery of quality services.[19] The current federal administration aims

**Table 1** Primary healthcare (PHC) context in India, Mexico and Uganda

| | India | Mexico | Uganda |
|---|---|---|---|
| Population, 2020 | 1.38 billion | 128 million | 45 million |
| GDP per capita, PPP (current international $), 2019[26] | 6996.56 | 20 944.03 | 2284.27 |
| Life expectancy at birth (years), 2019[27] | 70.8 | 76.0 | 66.7 |
| Maternal mortality ratio (per 100 000 live births), 2017[28] | 145 | 33 | 375 |
| Under-five mortality rate (deaths per 1000 live births), 2020[28] | 35.7 | 16 | 57.1 |
| Organisation | Three tired system:<br>► Subcentre the most peripheral and first contact point between the community and health system<br>► PHC is the first contact point between village community and the Medical Officer<br>► Community Health Centre with specialised medical and paramedical staff is the referral unit for PHCs<br>► Tertiary level includes hospital and medical colleges | Three level public health system in health districts:<br>► Community health centres with a medical doctor student in social service and a nurse<br>► Integrated Community Health Centres with health personnel, nurses, medical doctors, nutritionist, physical activator, social worker<br>► Secondary level includes general hospitals and staff of medical specialties is the referral unit for CHCs<br>► Tertiary level includes hospital of high specialties | Five-level system with health centres I, II, III, IV and the general hospital being the apex of the PHC system. All this functions with the Health Sub District administrative system |
| Financing | ► In 2015–2016, 43% of out of pocket expenditure by households was done on primary care.[29]<br>► National Health Policy 2017 commits a major proportion (>2/3rds) of resources to PHC | ► In 2018, 50% of total health spending came from Government schemes and compulsory contributory healthcare financing schemes, of which 24% was spent on primary care units.<br>► 42% of total health spending was out-of-pocket<br>► Population with household expenditures on health greater than 10% of total household expenditure or income (SDG indicator 3.8.2) 1.5% | Uganda's out of pocket on primary care increased through 38.4% through a period of 2004–2018.<br>The government of Uganda expenditure on health has stagnated at around 9.6% of its GDP with regards to the Abuja declaration of 15% |
| CPHC | The Health and Wellness Centre (HWC) component of Ayushman Bharat Programme aims to provide CPHC by upgrading and making 150 000 existing subcentres and primary health centres functional by December 2022.[18] The first HWC was launched on 14 April 2018 and by 31 March 2020, a total 38 595 AB-HWCs were operational across India.[30] | By 2018, 19% of population have no Universal Health Coverage,[31] and to solve this, The National Health Plan 2018–2024 create the Health Institute for Welfare, component of Mexican Health System, and aims to provide CPHC by organising health districts based in geographical areas by 2024. | The Uganda National Minimum Healthcare Package comprises of interventions that address major causes of morbidity and mortality both communicable and non-Communicable diseases including disease prevention and health promotion. This package of services is funded by government |

CPHC, comprehensive primary healthcare; PPP, Purchasing Power Parity; SDG, Sustainable Development Goal.

to strengthen the national health system through a 6-year Sectorial Health Programme (2019–2024). The pillars of the transformation are universal access to health services and free medicines for the entire population, a new CPHC-I model, the reorganisation of the health system moving from decentralised to a centralised system, the strengthening of the national pharmaceutical industry, and promotion of research.

PHC system in Uganda: Uganda started implementing health sector reforms in the late 1980s and early 1990s as part of a broader decentralisation policy to restore the health system after the political crises of 1970s. Decentralisation allowed the district authorities to cater to the local needs of the communities in terms of service delivery and strategic planning.[20] This decentralised system is based on the district as an administrative unit,

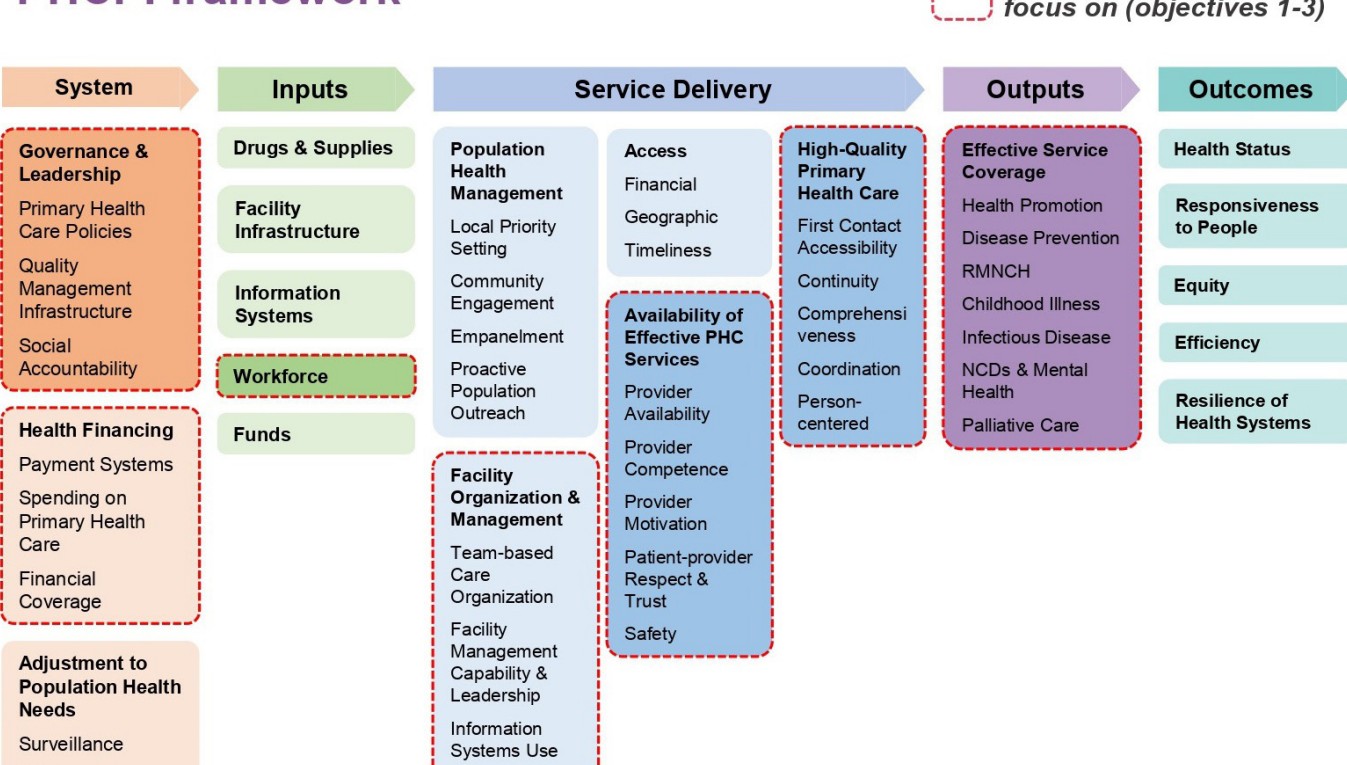

**Figure 1** Primary healthcare performance initiative (PHCPI) conceptual framework. NCD, Non Communicable Disease; PHC, primary healthcare; RMNCH, Reproductive, Maternal, Newborn and Child Health.

with the local government providing stewardship. PHC follows this decentralised system with multilayered healthcare delivery from health centre levels 1–4, and the general hospital at the apex. PHC administration is based on a Health Sub-District (HSD) system. Each HSD oversees several lower-level health facilities and provides supportive supervision. PHC is provided by nurses, clinical officers and non-specialist doctors, referred to as medical officers. This PHC approach links with the community through the Village Health Teams, which includes non-trained community members. Each health facility also has community members as members of the health unit management committee as a way of involving the community in the management and delivery of the health services. The aim of the HSD is to improve quality of routine health service delivery, increase equity of access to essential health services and foster community involvement in planning, management and delivery of healthcare.[21]

## METHODS
### Conceptual framework
The study will use the Primary Health Care Performance Initiative (PHCPI) conceptual framework (figure 1), and the research will be based on the service delivery and

output domains (availability of effective PHC services and high-quality PHC, effective service coverage) with a specific focus on the relationship between comprehensiveness of PHC (one of the key quality related PHC issues) and the composition (availability of groups of PHC providers with diverse education and capabilities) and organisation (team-based organisation of care to leverage the distinct expertise of different groups for provision of comprehensive PHC) of PHC teams, and to compare models between countries.[22 23]

### Study design
This will be a parallel mixed-methods study, which will combine qualitative and quantitative data in each country and support cross-country comparisons. Empirical data will be collected from PHC settings in a prespecified region from the three countries. It will comprise three steps as shown in figure 2.

### Patient and public involvement
Patients or public were not involved in the design, conduct or reporting or dissemination of this protocol.

### Site selection
Region and site selection will occur purposively to capture the diversity and needs of the population. Each country

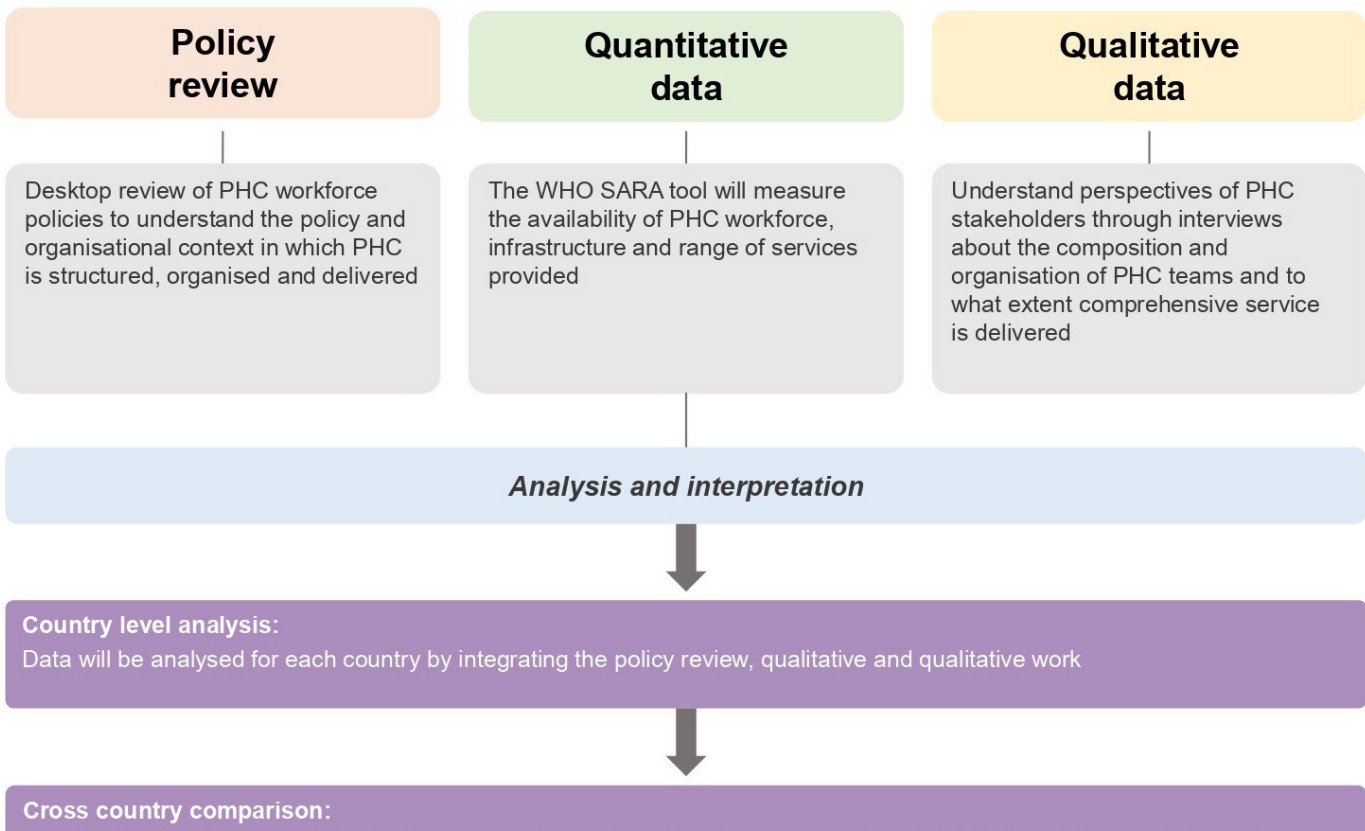

**Figure 2** Mixed-methods study design. LMIC, low-income and middle-income country; PHC, primary healthcare; SARA, Service Availability and Readiness Assessment.

will first select the regions, and then sites to represent the health needs and overall health system performance of the regions. Overall, 10 health units will be selected for each country except for India where 20 health units will be selected representing the regions, giving a total of 40 PHC units.

In India, a total of 20 PHCs from two regions (Vizianagaram from Andhra Pradesh, South India and Jhajjar from Haryana, North India) have been chosen. In Mexico, 10 PHCs will be included (three PHCs from Northern region, two from Western region; three from Central region and finally two PHCs from South region). In Uganda, 10 PHCs will be selected from the Eastern (Tororo district) and Western (Buliisa district) regions and will include one general hospital (figure 3).

### Data collection and analysis
#### Objective 1
To review national and subnational policies on PHC team composition and organisation and expected comprehensiveness of PHC.

#### Data collection
A desktop review of published and grey literature documents as well as relevant policy documents will be

conducted to identify the government regulations or policies related to PHC workforce.

#### Analysis
We will review the policies relating to PHC workforce and extract data to a standardised data collection tool template that uses the PHCPI conceptual framework (figure 1) with the below mentioned categories. (1) Governance and leadership; (2) Government spending on PHC; (3) PHC structure and organisation; (4) PHC workforce; (5) PHC service delivery and (6) PHC performance. Data will then be analysed qualitatively using NVivo software to create a narrative synthesis of the country's policy on the areas of interest.

#### Objective 2
Describe the actual composition and organisation of PHC teams.

#### Data collection
A cross-sectional descriptive survey will be conducted in the selected PHC facilities. Data collectors will be trained in WHO's Service Availability and Readiness Assessment (SARA) tool and will complete the questionnaire using electronic devices. SARA is a health facility assessment

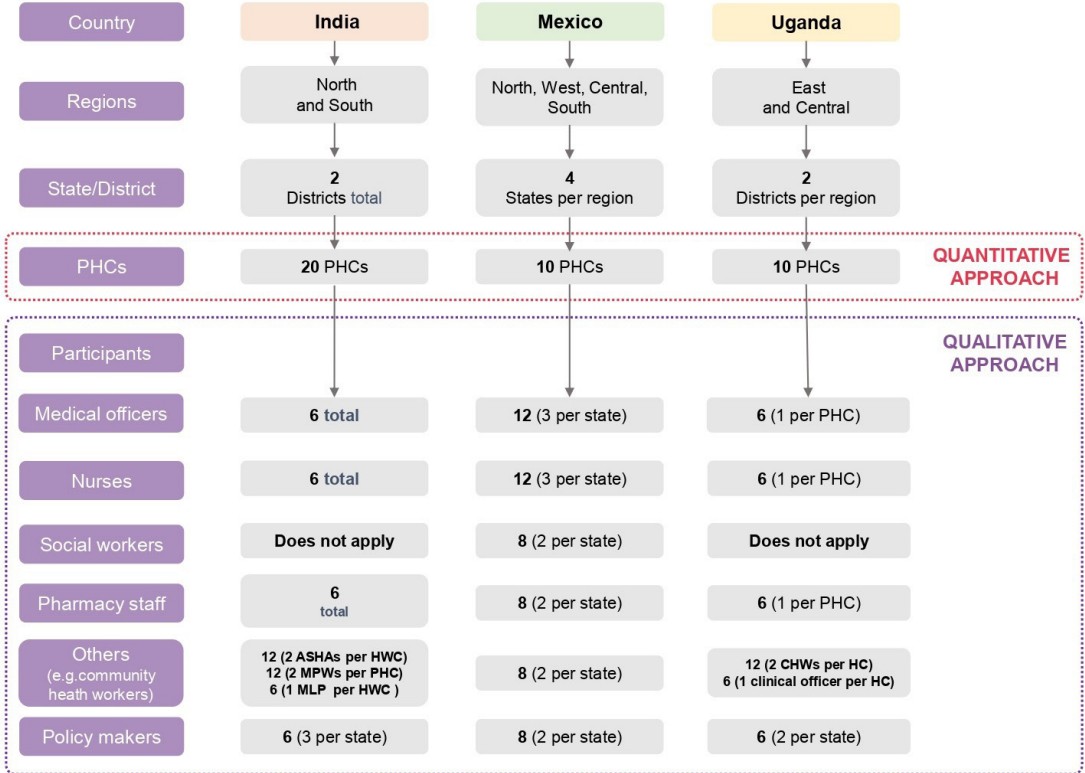

**Figure 3** Sample level distribution and methodological approach. HC, Health Care; HWC, Health and Wellness Centre; MLP, Mid Level Provider; MPW, Multipurpose Health Worker; PHC, primary healthcare.

tool designed to assess the available infrastructure, equipment and workforce, thereby determining the service availability and readiness of the facility to provide CPHC. We will not collect information about the availability of medicines at the PHC level as this study is focusing on health workforce. Data collection will occur at the PHC unit including its community-based outreach centres (eg, Health and Wellness Centres in India, at the selected PHC units in Mexico and health centres 2 and 3 in Uganda) to understand the PHC infrastructure, composition of PHC teams and the services delivered to the community. Data will be collected on electronic devices using the Open Data Kit platform, stored locally on the device, and when internet connectivity is available, uploaded to a central repository/server in respective countries for data analysis. When internet is not available, data from the devices can be manually saved in the central repository.

## Analysis

Service availability will be described by three domains: health infrastructure, health workforce and service utilisation. Continuous variables will be summarised using either mean (SD) or median (IQR). All categorical variables will be summarised using frequencies and percentages.

## Objective 3

Assess the comprehensiveness of care provided by PHC teams.

## Data collection

This comprises semistructured in-depth interviews (IDIs) to explore topics on the role and recruitment of the workforce, and how jobs are shared in the team, training, accreditation, supervision, performance evaluation, incentives, career progression, community involvement, team composition, organisation and comprehensiveness of services provided. Comprehensiveness of services will be assessed by asking which services are delivered, the range of conditions addressed by the team, if the workforce is trained in managing those conditions, and do the range of services include prevention, promotion, treatment, rehabilitation and palliation? For instance, does that PHC provide care for cardiovascular risk factors and if so, are the staff trained and do they have access to the necessary equipment to measure the risk factors? A purposive sample of participants including PHC workforce (community health workers, nurses, social workers, pharmacy staff, health promoters, primary care doctors), and National/Regional/District level policy makers and PHC managers will be invited for the IDIs. Trained researchers from each country will interview participants in local languages (Telugu, Hindi and English in India; Spanish in Mexico; and English, Ateso, Jopadhola and Runyoro in Uganda) using interview guides described in Appendix 1, 2 and 3. Debriefing sessions with the entire research team will be held each week. Interviews will take place over phone/zoom/skype or in-person depending on the local situation of COVID-19 pandemic and will

be audiorecorded. Participants will be contacted at the health units or their office (policy makers) and will be interviewed in an area within the unit that meets the appropriate privacy conditions. We aim to conduct up to 60 interviews in each country (180 interviews in total) (see online supplemental files 1–3).

## Data analysis

Interviews will be transcribed verbatim in-country and transcripts in Hindi, Telugu, Spanish and Ateso, Jopadhola and Runyoro will be translated to English for analysis. The qualitative data for each country will be coded using NVivo software (QRS International, Vic) and analysed using an inductive approach. Two coders from each country will review and analyse the data. Weekly calls will be set up to discuss the emerging themes with the research team. This approach will enable us to explore and identify the important issues in PHC workforce organisation, composition and comprehensiveness, and will also help us to identify shared challenges and differences across countries.

### Triangulation of data

The emergent themes from the qualitative interviews in each country will be interpreted in conjunction with the SARA survey and outputs from the policy analysis. Data integration of the three objectives will help us identify the policy and implementation gaps for each country.

### Objective 4

Conduct a comparative analysis of the relationship between PHC team composition and organisation with the delivery of comprehensive PHC across the three countries.

### Data analysis

We will use a case-oriented research strategy where each 'case' (country) will be considered analytically as a whole.[24 25] Comprehensiveness of services (which services such as prevention, promotion, treatment, rehabilitation or palliation; for what conditions and by whom) will be explored through the policy review, SARA (availability of infrastructure to deliver CPHC) and interviews with PHC team members. Cross country comparisons will be conducted to understand similarities and differences in PHC-related policies, especially in terms of the workforce composition, organisation and service delivery with the intention of learning about the different approaches to CPHC and PHC workforce organisation, the context in which PHC systems exist, and why they take the forms they do. The comparison will examine the differences and similarities between PHC policies, organisation and service delivery in the three countries.[25]

## ETHICS

Ethical approvals have been sought from Institutional Ethics Committee of The George Institute for Global Health, India for the Indian sites (Ref 16/2020); School of Medicine Research Ethics Committee at Makerere University for the sites in Uganda (Ref 2020-218); and the Mexican National Institute of Public Health (INSP for its Spanish acronym) ethics review board for the sites in Mexico (Ref: 1726). Additional permissions have been sought from the Uganda National Council for Science and Technology and Tororo and Buliisa District Health Offices and the INSP Research committee. The local health authorities will provide approval for collection of data at the facility level. The respondents will be adequately informed regarding all relevant aspects of the study, including its aim and interview procedures, through a participant information sheet. Respondents who accept to participate in the study will provide signed written informed consent. All participants will be given written participant information sheets prior to consenting to participate in this study. Data collection instruments will be piloted and administered by means of electronic questionnaires on mobile devices. We anticipate that data collection for SARA and semistructured interviews will take approximately 12 months and analysis for the entire study will take additional 6 months. As data collection is taking place during the COVID-19 pandemic, provision to conduct interviews online has been made accordingly.

## SIGNIFICANCE

This study will provide insight into the availability of policies on PHC, the implementation of policies on PHC workforce, team organisation and service provision for the delivery of CPHC. Furthermore, it will investigate how different LMICs organise their PHC teams to deliver UHC through comprehensive primary care.

**Author affiliations**

[1]Global Health, School of Population Health, University of New South Wales, Sydney, New South Wales, Australia
[2]Better Care, The George Institute for Global Health, New Delhi, India
[3]Makerere University, Kampala, Uganda
[4]Centro de Investigación en Sistemas de Salud, INSP, Cuernavaca, Morelos, Mexico
[5]Health Systems Science, The George Institute for Global Health, Sydney, New South Wales, Australia
[6]Faculty of Medicine, UNSW, Sydney, New South Wales, Australia
[7]Family Medicine and Primary Care, Stellenbosch University, Cape Town, Western Cape, South Africa
[8]Centro de Investigación en Sistemas de Salud, Instituto Nacional de Salud Pública, Cuernavaca, Mexico
[9]General Practice and Primary Health Care, University of Auckland, Auckland, New Zealand
[10]UNICEF, Vijayanagram, Andhra Pradesh, India
[11]Post Graduate Institute of Medical Science, Rohtak, Haryana, India
[12]Research and Development, The George Institute for Global Health, Hyderabad, Telangana, India

**Acknowledgements** We would like to acknowledge the Primary Healthcare Research Consortium and the team in Uganda (Midiam Ibáñez-Cuevas, Carlos Chivardi-Moreno, Namatovu Jane, and Onyango Jude) for their contribution. This work was supported, in whole, by the Bill & Melinda Gates Foundation (INV-000970). Under the grant conditions of the Foundation, a Creative Commons

Attribution 4.0 Generic License has already been assigned to the Author Accepted Manuscript version that might arise from this submission.

**Contributors** The study was designed by RJ, DPr and DPe. The first draft was written by RJ, IB, DPe with inputs from RJM, FG-S and DPr. MS, IB, HR, RJo, ES-M, LA-B, EO, DVO-A, NJ, OJ, RJe, SU, VA, DPe provided inputs to the protocol. All authors have read and approved the final manuscript.

**Funding** This study is funded by the Bill & Melinda Gates Foundation through the Primary Healthcare Research Consortium hosted at the George Institute for Global Health in India. RJ is funded by the National Heart Foundation of Australia (Grant number 102059) and the University of New South Wales Scientia Fellowship (No grant number).

**Competing interests** None declared.

**Patient and public involvement** Patients and/or the public were not involved in the design, or conduct, or reporting, or dissemination plans of this research.

**Patient consent for publication** Not applicable.

**Provenance and peer review** Not commissioned; externally peer reviewed.

**ORCID iDs**
Rohina Joshi http://orcid.org/0000-0002-3374-401X
Ileana Heredia-Pi http://orcid.org/0000-0002-9998-9239
David Peiris http://orcid.org/0000-0002-6898-3870
Robert James Mash http://orcid.org/0000-0001-7373-0774
Hortensia Reyes-Morales http://orcid.org/0000-0002-9763-4143
Felicity Goodyear-Smith http://orcid.org/0000-0002-6657-9401
Renu John http://orcid.org/0000-0001-9652-034X
Doris V Ortega-Altamirano http://orcid.org/0000-0003-4767-8268
Emanuel Orozco-Núñez http://orcid.org/0000-0002-6550-7385

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
