## [Reviewer comments · BMJ Open]

ARTICLE DETAILS

TITLE (PROVISIONAL)	How do diverse low- and middle-income countries implement primary health care team integration to support the delivery of comprehensive primary health care? A mixed methods study protocol from India, Mexico and Uganda
AUTHORS	Joshi, Rohina; Besigye, Innocent; Heredia, Ileana; Sharma, Manushi; Peiris, David; Mash, Robert; Reyes, Hortensia; Goodyear-Smith, Felicity; John, Renu; Ortega-Altamirano, Doris; Orozco, Emanuel; Ávila-Burgos, Leticia; Jeyakumar, Ragavi; Serván-Mori, Edson; Upadhyaya, Sanjeev; Arora, Varun; Praveen, D

VERSION 1 – REVIEW

REVIEWER	Kawade, Anand King Edward Memorial Hospital, Vadu Rural Health Progm
REVIEW RETURNED	14-Sep-2021

GENERAL COMMENTS	Thank you for the opportunity to review this manuscript. At the outset I would like to congratulate the authors for choosing this important research area. The results of this study will help to address the knowledge gaps in PHCs. Global evidence suggests that to achieve universal health coverage and SDG, high quality primary health care which is people centered, responsive, accessible, comprehensive, adaptive, affordable and continuous is essential. Current Covid-19 pandemic have underscored this need. The findings of this study will help health systems of different countries to consider renovating their PHCs and CPHC. Comments: Abstract: Line 12: add “PHC team integration, composition and organization” in aim Ethics and Approvals: As you are conducting the study across the health system, are you seeking permission and approval from health systems? In data collection and analysis: For objective 3: you have described more of PHC team rather than comprehensiveness of care. Please add how you are collecting the data on comprehensiveness of care and how you are planning to analyze these data. Methods: 1) I didn't find any timelines of this study. Having a qualitative component in it, we expect that this qualitative part will require substantial time. Therefore, we request authors to state the timelines (preparatory, tool development, approvals, data collection, analysis etc.)
--

	2) I am not clear whether authors are using complete WHO's SARA tool or restricting to questions related to human workforce to address our objective of PHC team integration & organization. This is very important for convergence of qualitative and quantitative data. Please clarify the same. 3) Again, I am not clear whether authors are modifying the SARA tool as per the country's health system characteristics? Needs clarification. 4) Before interviewing the key informants, please clarify the selection criteria for these key informants like age, education, gender, experience etc. because the quality of information will depend on these factors. 5) In interviewer guides of policy makers, it would be better if you can add a question about their perception about the size and composition of PHC team. 6) I am not sure whether we use term "convergent mixed methods". To my understanding when we triangulate and connect the qualitative & quantitative data, we could come across the divergent themes which are indicative of future research areas, hence requires mention. So, we can use simply "mixed methods". Thank you once again and all the best !!
--	--

REVIEWER	Sevdalis, Nick King's College London
REVIEW RETURNED	20-Dec-2021

GENERAL COMMENTS	The authors report a study protocol for a mixed methods study across three low and middle income countries. The focus of the study is on primary health care provision, specifically from the perspective of how such care systems are designed and their workforce, incl. what personnel delivers primary health care on the ground, their skills, training and supervision. The study design is prospective and will include a comparative case study element, where the data from the three countries (India, Mexico and Uganda) will be compared and contrasted. The datasets will include policy documentation and will rely to a large extent on semi-structured interviews with people working within primary care and community-based facilities in the study countries. Ethical review is adequately covered. This is a funded study to be delivered by a global team. The rationale and need for the study as well as its aims and objectives have been made clear and make sense. The proposed methods seem to me to be suitable to address the aims and they have been well-reported – the visual materials in particular are very helpful. I have no reservations regarding this protocol or any further recommendations for the authors. Good luck with the data collection.
--

VERSION 1 – AUTHOR RESPONSE

Reviewer: 1

1. Abstract: Line 12: add “PHC team integration, composition and organization” in aims.
Thank you. This has been added. “This study aims to understand PHC team integration, composition and organization in the delivery of CPHC in India, Mexico and Uganda.”

2. Ethics Approvals: As you are conducting the study across the health system, are you seeking permission and approval from health systems?

The study was discussed in detail with the sub-national policy makers and the health system personnel. Appropriate approval was obtained from local offices to conduct the study. Overall, the ethics approval for the study was obtained from an independent Ethics Committee based at the George Institute for Global Health in India. This study has been registered with the Clinical Trials Registry - India (REF/2021/05/043528) and is applicable for the overall study

3. In data collection and analysis: For objective 3: you have described more of PHC team rather than comprehensiveness of care. Please add how you are collecting the data on comprehensiveness of care and how you are planning to analyze these data.

We have included lines 1-4 on page 11 and lines 2-5 on page 12 to explain how data for comprehensiveness will be collected and analysed.

“Comprehensiveness of services will be assessed by asking which services are routinely delivered, the range of conditions addressed by the team and if the team is trained in managing those conditions. Do the range of services include prevention, promotion, treatment, rehabilitation and palliation? For instance, does that PHC provide care for cardiovascular risk factors and if so, are the staff trained and do they have access to the necessary equipment to measure the risk factors?”

“Comprehensiveness of services (which services such as prevention, promotion, treatment, rehabilitation or palliation; for what conditions and by whom) will be explored through the policy review, SARA (availability of infrastructure to deliver CPHC) and interviews with PHC team members.”

4. I didn't find any timelines of this study. Having a qualitative component in it, we expect that this qualitative part will require substantial time. Therefore, we request authors to state the timelines (preparatory, tool development, approvals, data collection, analysis etc.)

We have added information about timelines on page 13 (lines 2-5)

“We anticipate that data collection for SARA and semi-structured interviews will take approximately 12 months and analysis for the entire study will take additional six months. As data collection is taking place during the COVID-19 pandemic, a provision to conduct interviews online has been made accordingly.”

5. I am not clear whether authors are using complete WHO's SARA tool or restricting to questions related to human workforce to address our objective of PHC team integration & organization. This is very important for convergence of qualitative and quantitative data. Please clarify the same. Again, I am not clear whether authors are modifying the SARA tool as per the country's health system characteristics? Needs clarification.

Thank you for this comment. We did not modify the SARA tool, however we did not collect information about medicines as this study focussed on workforce comprehension and organisation. We have indicated this in the paper. Page 10, lines 15 and 16.

“We will not collect information about the availability of medicines at the PHC level as this study is focussing on health workforce.”

6. Before interviewing the key informants, please clarify the selection criteria for these key informants like age, education, gender, experience etc. because the quality of information will depend on these factors.

Participants from all PHCs which were sampled for this study were invited for the interviews. This included PHC workforce (community health workers, nurses, social workers, pharmacy staff, health

promoters, primary care doctors), and National/Regional/District level policy makers and PHC managers. We did not have additional requirements. (Please see page 11, lines 10-20)

7. In interviewer guides of policy makers, it would be better if you can add a question about their perception about the size and composition of PHC team.

Thank you for this suggestion.

8. I am not sure whether we use term “convergent mixed methods”. To my understanding when we triangulate and connect the qualitative & quantitative data, we could come across the divergent themes which are indicative of future research areas, hence requires mention. So, we can use simply “mixed methods”.

Thank you, we have changed this to ‘mixed-methods’ on pages 3 and 9.

Reviewer: 2

1. I have no reservations regarding this protocol or any further recommendations for the authors.

Good luck with the data collection.

Thank you

VERSION 2 – REVIEW

REVIEWER	Kawade, Anand King Edward Memorial Hospital, Vadu Rural Health Program
REVIEW RETURNED	28-Feb-2022
GENERAL COMMENTS	Thank you very much for addressing the comments to satisfaction. No further comments .All the best for data collection